# Finding the Way to Improve Motor Recovery of Patients with Spinal Cord Lesions: A Case-Control Pilot Study on a Novel Neuromodulation Approach

**DOI:** 10.3390/brainsci12010119

**Published:** 2022-01-17

**Authors:** Antonino Naro, Luana Billeri, Tina Balletta, Paola Lauria, Maria Pia Onesta, Rocco Salvatore Calabrò

**Affiliations:** 1IRCCS Centro Neurolesi Bonino Pulejo Piemonte, Via Palermo, SS 113, Ctr. Casazza, 98124 Messina, Italy; g.naro11@alice.it (A.N.); luanabilleri@hotmail.it (L.B.); tina.balletta@irccsme.it (T.B.); paola.lauria@irccsme.it (P.L.); 2Spinal Cord Unit, Cannizzaro Hospital, 95126 Catania, Italy; mp.onesta@virgilio.it

**Keywords:** gait motor function, neuroplasticity, direct current stimulation, spinal cord injury, robot-aided gait training

## Abstract

Robot-assisted rehabilitation (RAR) and non-invasive brain stimulation (NIBS) are interventions that, both individually and combined, can significantly enhance motor performance after spinal cord injury (SCI). We sought to determine whether repetitive transcranial magnetic stimulation (rTMS) combined with active transvertebral direct current stimulation (tvDCS) (namely, NIBS) in association with RAR (RAR + NIBS) improves lower extremity motor function more than RAR alone in subjects with motor incomplete SCI (iSCI). Fifteen adults with iSCI received one daily session of RAR+NIBS in the early afternoon, six sessions weekly, for eight consecutive weeks. Outcome measures included the 6 min walk test (6MWT), the 10 m walk test (10MWT), the timed up and go (TUG) to test mobility and balance, the Walking Index for Spinal Cord Injury (WISCI II), the Functional Independence Measure-Locomotion (FIM-L), the manual muscle testing for lower extremity motor score (LEMS), the modified Ashworth scale for lower limbs (MAS), and the visual analog scale (VAS) for pain. The data of these subjects were compared with those of 20 individuals matched for clinical and demographic features who previously received the same amount or RAR without NIBS (RAR − NIBS). All patients completed the trial, and none reported any side effects either during or following the training. The 10MWT improved in both groups, but the increase was significantly greater following RAR + NIBS than RAR − NIBS. The same occurred for the FIM-L, LEMS, and WISCI II. No significant differences were appreciable concerning the 6MWT and TUG. Conversely, RAR − NIBS outperformed RAR + NIBS regarding the MAS and VAS. Pairing tvDCS with rTMS during RAR can improve lower extremity motor function more than RAR alone can do. Future research with a larger sample size is recommended to determine longer-term effects on motor function and activities of daily living.

## 1. Introduction

Spinal cord injury (SCI) affects between 250,000 and 500,000 persons annually worldwide [1], often causing severe and permanent loss of motor, sensory, or autonomic functions. In addition, SCI has a striking socio-economic impact, as it often affects young people of working age. SCI thus requires an intensive rehabilitative approach to counteract the residual functional impairment [2].

Motor recovery occurs mainly within the first two months after SCI. However, chronic SCI patients may also have chances to recover further motor function with adequate, intensive training [3]. In this regard, robotic rehabilitation devices have been increasingly utilized as an adjunct therapy to the conventional rehabilitation strategies for individuals with SCI [4]. The rationale of adopting robot-aided rehabilitation (RAR) consists of enhancing motor function recovery through highly repeated functional movements and the entrainment of residual neural plasticity mechanisms subtending functional recovery, at either the spinal central pattern generator [5] or cortical level [6,7]. Furthermore, RAR allows counteracting the numerous constraints in providing an individualized training strategy, including reduced sensorimotor coordination, spasticity, and impaired balance [8]. Finally, RAR serves as a mobility aid beyond orthoses and wheelchairs [4].

Many promising RAR interventions have been shown to improve the mobility, function, and quality of life of individuals with SCI, in particular regarding lower extremity robotic exoskeletons. However, the majority of the available works have methodological and rehabilitation paradigm differences, thus being unable to demonstrate the superiority of one gait training strategy over another in counteracting the loss of muscle strength and trophism, walking disability and mobility, sensory dysfunction, autonomic disorders, spasticity, pain, and overall quality of life [9,10,11,12,13,14,15]. Therefore, more extensive studies are required to prove RAR’s benefits definitively [4].

Similar to RAR research in SCI patients, more innovative ways to stimulate the brain and spinal plasticity to promote functional recovery have been investigated. Mainly, non-invasive brain stimulation (NIBS) has been adopted to potentiate the therapeutic benefits of RAR [16]. The rationale of coupling NIBS with RAR mainly stems from the possibility to couple bottom-up (RAR) and top-down (NIBS) plasticity processes, thus better targeting the neural pathways that are responsible for motor (re)learning processes and are entrained during the rehabilitation processes owing to intensive, repetitive, assisted-as-needed, and task-oriented approaches [6,16,17,18]. However, significant concerns remain about NIBS’s administration time, order effect, and blinding when coupled to RAR. Therefore, further investigations are required to better assess the effects of this paired approach on motor function recovery following SCI, as demonstrated in patients with stroke [19,20].

The study aimed at ascertaining whether NIBS paired with RAR could provide SCI patients with superior outcomes related to gait, spasticity, and pain than stand-alone RAR. Given that we focused on the potential efficacy of a combined training with RAR and NIBS on gait, we considered only ambulatory individuals (i.e., incomplete SCI-iSCI- with ASIA C–D). Moreover, we recruited only chronic iSCI patients (i.e., more than 6 mo post-injury) so as to have individuals with a stable level of recovery, which could suggest that the observed improvements may depend on the intervention itself rather than on a spontaneous recovery [21], and without medical conditions that could preclude RAR utilization.

## 2. Materials and Methods

### 2.1. Participants and Study Design

We consecutively enrolled patients with SCI attending our Neurorobotic Unit between January 2016 and December 2019. Inclusion criteria were: (i) chronic (onset more than 6 mo), non-progressive (traumatic or non-traumatic), thoracic (between T3 and T10) iSCI (classified by the ASIA Impairment Scale (AIS) as grades C and D at entry); (ii) age range 18–65 y. In addition, pressure ulcers, severe range of motion limitation due to spasticity or tendon retraction, severe bone, heart, or pulmonary disease, and NIBS contraindication (e.g., implanted electromechanical devices) represented the exclusion criteria.

We screened 35 patients, 15 of whom were included in the study. The patients were evaluated at baseline (T0) using clinical scales as outcome measures. Then, patients were provided with a NIBS session followed by a RAR session. Each subject was provided with a daily session of RAR + NIBS in the early afternoon (six sessions weekly), for eight consecutive weeks. The patients were then evaluated immediately after (T1) and three months after (T2) the training, using the above clinical scales. The data of these subjects were compared with those coming from a sample of 20 individuals matched for clinical and demographic features who previously underwent the same amount or RAR without NIBS (RAR − NIBS). The clinico-demographic characteristics of both groups are summarized in Table 1.

### 2.2. Outcome Measures

We used the 6 min walk test (6MWT) and the 10 m walk test (10MWT) to measure ambulatory ability and endurance [22,23,24]. The timed up and go (TUG) allows testing mobility and balance [25,26]. The Walking Index for Spinal Cord Injury (WISCI II; scoring from 1 to 20) addresses the amount of physical assistance, braces, or devices required to walk 10 m [27]. The Functional Independence Measure-Locomotion (FIM-L) quantifies the need for assistance when performing physical, psychological, and social functions [28,29]. The manual muscle testing for lower extremity motor score (LEMS; five key muscles—hip flexors, knee extensors, ankle dorsiflexor, long toe extensors, and ankle plantar flexors—of both lower extremities; range 0 to 50) was conducted according to the ASIA standard. Spasticity was assessed using the modified Ashworth scale (MAS) for lower limbs. The visual analog scale (VAS) was used to quantify individual estimation of pain [30].

### 2.3. Non-Invasive Brain Stimulation

NIBS consisted of a repeated transcranial magnetic stimulation (rTMS) paradigm carried out simultaneously with a cathodal transvertebral direct current stimulation (tvDCS) (also known as transcutaneous spinal direct current stimulation) paradigm.

rTMS was delivered using a Magstim Super-Rapid2 stimulator (Magstim Company, Whitland, U.K.) equipped with a double-cone coil (each wing measuring 110 mm in diameter) perpendicularly held over the vertex, in order to trigger both legs’ primary motor areas. According to a brain MRI scan, the stimulation site was identified and marked on a personal head-cap. Coil positioning was carefully documented (including position, angulation, and inclination) to keep it constant along with rTMS sessions. The coil was held in position by a mechanical support. The stimulation intensity was set at 90% of the right tibialis anterior muscle resting motor threshold (RMT). Each session consisted of 60 bursts of 20 pulses at 10 Hz with inter-train intervals of 10 s, for 1200 pulses.

tvDCS was delivered using a Brain Stim device (EMS; Bologna, Italy) equipped with two rubber electrodes of 49 cm^2^ inserted in a saline-soaked sponge, which were fixed over two metameres above the site of spinal lesion, serving as the active electrode, and over the left deltoid, serving as the reference electrode. Skin impedance was adequately reduced using abrasive gel and then wiped clean with alcohol swabs. The stimulation intensity was set at 2 mA and lasted 20 min (current density of 0.041 mA/cm^2^ and charge density of 0.048 C/cm^2^). The current was ramped up to the full intensity over 30 s at the onset of tvDCS and ramped down over 30 s at the end.

### 2.4. Robot-Aided Rehabilitation

RAR consisted of a neurorobotic treatment using the LokomatPro (i.e., a Lokomat with an Augmented Performance Feedback) (Hocoma; Volketswil, Switzerland). Lokomat is a robotic device consisting of powered gait orthoses with integrated computer-controlled linear actuators at each hip and knee joint, a body-weight support system (BWSS), and a treadmill [31]. The augmented performance feedback guarantees motivating, challenging, and instructive functional feedback in virtual environments. Patients performed a forty-minute session per day, in the early afternoon, from Monday to Friday, for eight consecutive weeks, for a total of forty sessions. The amount of BWS was initially set at 70% of the patient’s weight, then decreased according to the patient’s load tolerance, and the gait speed was adjusted to make the exercise comfortable for the patient. A Lokomat-trained physiotherapist supervised each session. In addition to RAR sessions, patients underwent conventional physical therapy (CPT) twice a day and five-times a week using the Bobath principles, occupational therapy, and functional electrical stimulation.

### 2.5. Statistical Analysis

The primary analysis sought the changes of the 6MWT, 10MWT, TUG, and WISCI II (all gait-related outcome measures) in the two groups from the baseline over the treatment period (T1 vs. T0 and T2 vs. T0). In this regard, ANCOVA was used, adjusting for the baseline value. This analysis was also performed for all the other outcome measures.

The secondary analysis sought the changes from the baseline over the treatment period (T1 vs. T0 and T2 vs. T0) of the outcome measures depending on the baseline ASIA scorings using a two-way ANOVA with *group* (2 levels: RAR + NIBS and RAR − NIBS) and *time* (3 levels: T0, T1, and T2) as factors. Pairwise comparisons with Bonferroni correction were tested.

Finally, we categorized the subjects as improved (WISCI II changes superior to the MCID) or not improved (WISCI II changes non-superior to the MCID) to identify possible predictors of recovery after rehabilitation. To this end, we used a multivariable logistic regression with the clinico-demographic features at the baseline (age, gender, time since SCI, NLI, etiology, AIS score, presence of spasticity, presence of pain, and LEMS value) as candidate predictors.

All the analyses were conducted according to an intention-to-treat analysis, thus including all participants for which data were available. The significance level of the statistical data was set at α < 0.05.

We estimated that 64 individuals should be studied assuming 80% power, a type I error of 0.05, a mean difference of 30% on the WISCI II, and a dropout rate of 10%.

The experimenters who analyzed the data were blinded to the patients’ allocation.

## 3. Results

There were no significant differences in the clinico-demographic features and treatment periods between the groups. Most participants were taking medications (as shown in Table 1). However, all participants were hospitalized at the Neurorobotic Unit of our Institute so that the patients’ medication status was easily controlled. Further, patients were not provided with any medication change during the experimental period.

All patients completed the trial, and none reported any side effects during or after the training. The 10MWT improved in both groups, but the increase was more significant following RAR + NIBS than RAR − NIBS (Table 2). The same occurred for the FIM-L, LEMS, and WISCI II. No significant differences were appreciable concerning the 6MWT and TUG. Conversely, RAR − NIBS outperformed RAR + NIBS concerning the MAS and VAS.

There was no significant effect of patients’ stratification depending on ASIA on clinical outcome measure changes (all *p* > 0.1).

The significant predictors of recovery were the LEMS, age, and time since injury (all *p* < 0.0001).

## 4. Discussion

RAR has been proven effective in post-SCI gait rehabilitation, as it can bypass the constraints in providing an individualized training strategy and the main limitations of iSCI individuals in overground walking ability, i.e., sensorimotor coordination, spasticity, impaired balance, and muscle weakness [8,32,33,34,35,36]. Consistently, the RAR − NIBS group showed a significant improvement in ambulation (including walking speed and independence and lower limb muscle strength) with a reduced requirement of assistance after the treatment. In addition, significant spasticity and pain reduction were also appreciable.

These effects could depend on the fact that RAR provides repetitive, intensive, assisted-as-needed, and task-oriented treatment, which can entrain the sensorimotor cortex and the cerebellar regions involved in gait control, thus leading to a motor performance improvement [37]. Furthermore, RAR provides proprioceptive inputs to the lower extremities that, consistent with gate control theory, block noxious small fiber afferents, which cause pain and spasticity, thus contributing to gait improvement [38,39].

Therefore, RAR combined with CPT offers some valuable clinical benefits. Contrarily, other studies showed no significant difference between RAR and CPT [9,10,11,12,40]. Therefore, further investigation is necessary to find additional strategies to RAR that may further promote rehabilitation outcome achievement.

NIBS has been shown to contain motor impairment and promote spinal fiber functional restoration [41,42]. However, to the best of our knowledge, only a few studies have investigated the feasibility and potential efficacy of RAR paired with NIBS on gait performance in individuals with iSCI [43,44,45,46]. Remarkably, no study assessed both rTMS and tvDCS contemporarily with RAR.

We found that RAR + NIBS was safe and feasible, as all patients completed the trial and none reported any side effects during or after the training. In addition, all patients were well compliant with the NIBS protocol. The RAR + NIBS group outperformed the RAR − NIBS group concerning gait speed, muscle strength, ambulation autonomy, and disability burden. Conversely, both groups significantly improved in gait endurance, balance, spasticity, and pain, without any between-group difference.

The combined approach could be thus preliminarily considered a training strategy to provide safe and more effective neuromuscular re-education for iSCI patients compared to RAR alone. Another advantage coming from the implementation of NIBS in the RAR strategy lies in its extensive applicability to SCI patients, as its aftereffects were independent of patients’ ASIA. However, this requires confirmation by randomized trials, as we adopted a propensity matching analysis for observational studies. Furthermore, some clinical features may critically influence outcome achievement, including baseline LEMS (the better the LEMS, the higher outcomes scores), patients’ age (the younger the patient, the higher the outcomes scores), and time since injury (the earlier the rehabilitation period begins, the higher the outcomes scores are). When selecting patients to be submitted to NIBS paradigms, these issues should be taken into account, but clarification and confirmation are needed from randomized clinical trials.

The neurophysiological underpinnings of RAR + NIBS are not clear. The paradigm we implemented, i.e., rTMS paired with tvDCS, is entirely new. Neurophysiological measures were not pursued in this study. Therefore, we can only hypothesize that this double-NIBS may have triggered both spinal and supraspinal mechanisms mediating N-methyl-D-aspartate (NMDA) receptor and gamma-aminobutyric acid (GABAergic) activity-mediated neuroplastic changes [47,48,49,50,51,52,53,54,55,56,57,58,59,60]. These mechanisms are also triggered by RAR [5]. Based on these issues and our data, the hypothesis that cathodal tvDCS paired with RAR could induce more evident changes in neuroplasticity and gait compared to RAR alone is plausible. Specifically, the efficacy of the coupled intervention may lie in a shared target by NIBS and RAR, i.e., the activity of spinal interneurons within the central pattern generator. In addition, both approaches may provide sufficient sensory–motor stimulation to optimize neural plasticity [5]. Therefore, a synergistic effect is hypothesized, despite the underlying neurophysiological mechanism remaining partially unclear [61]. These may consist of cell death limitation, regeneration and replacement, remyelination, and spinal plasticity mechanisms’ modulation [62]. Consistent with our previous findings in stroke models, we suggest that NIBS and RAR potentiate each other in a sort of paired associative stimulation, thanks to either direct cortico-spinal or trans-synaptic spinal effects [61,62,63,64,65].

### Limitations

Our study did not use randomization, which is the major study limitation. Randomized trials enable the unbiased estimation of treatment effects and imply that treatment groups are balanced on average for each covariate. Conversely, the propensity matching analysis of the subjects we adopted may have a limited strength of the beneficial effect of the experimental approach. In addition, there is a non-negligible possibility of bias due to a difference in the treatment outcomes (such as the average treatment effect) because a factor predicts treatment rather than the treatment itself. Unfortunately, for observational studies such as ours, treatment assignment to research subjects is typically not random. However, between-group patient matching (as we did in our study) attempts to reduce the treatment assignment bias and mimic randomization by making a sample of units that receive the treatment comparable to a sample of units that do not receive the treatment concerning all observed covariates. Furthermore, given the novelty of the current approach (combined neuromodulation strategies with RAR), we preferred to implement a pilot trial to examine the safety and feasibility of such a combined neuromodulation approach for iSCI patients, giving insights for further randomized trials. Finally, the propensity matching analysis we adopted also has some advantages, including the estimation of the covariates that predict receiving the treatment and the reduction of the biases due to confounding variables that could be found in an estimate of the treatment effect obtained from simply comparing outcomes among units that receive the treatment versus those that do not [66].

Other limitations of our study include the non-homogenous etiology (traumatic/non-traumatic), the short-term follow-up (up to three months), the lack of neurophysiological measures, the lack of control groups such as sham tvDCS, and tvDCS paired with CPT (only RAR without any NIBS was available). Furthermore, we included in our study only patients in the chronic phase. On the one hand, this limits the applicability of our approach; on the other hand, this sample selection was intended to avoid the bias related to spontaneous recovery, which is commonly observed in acute/subacute patients.

This notwithstanding, our study was intended to preliminarily focus on the safety, feasibility, and potential effectiveness of RAR + NIBS as compared to RAR alone and to provide basic information on determining the appropriateness of candidates and the optimal timing, to design the maximal efficacy of RAR in SCI patients for future randomized clinical trials.

## 5. Conclusions

Cathodal tvDCS paired with rTMS is promisingly safe, feasible, and effective in potentiating RAR plus CPT outcome achievement. Furthermore, RAR alone was confirmed as effective to improve function ambulation in motor iSCI. This combined approach could be considered an effective training strategy to provide safe and efficacious neuromuscular re-education for iSCI patients once these promising data are confirmed by more extensive randomized controlled trials incorporating objective clinical and neurophysiological measures of corticospinal and spinal excitability.

## Figures and Tables

**Table 1 brainsci-12-00119-t001:** Clinico-demographic characteristics (summarized as percentage or mean ± sd).

	Gender	Age (y)	TSO (m)	SCI Level	Etiology	AIS Score	Spasticity (Yes/No), Medication		Pain (Yes/No), Medication		
RAR + NIBS	M	35	6	8	V	D	yes	no medication	yes	paracetamol	
	M	44	6	9	T	D					
	F	37	6	9	T	C			yes	no medication	
	M	40	6	5	T	D			yes	no medication	
	F	48	6	6	T	D	yes	tizanidine	yes	paracetamol	
	F	46	8	3	T	D	yes	tizanidine			
	M	34	8	3	TM	C			yes	gabapentin	
	F	36	9	5	TM	C	yes	tizanidine	yes	gabapentin	
	M	22	10	9	V	C	yes	tizanidine	yes	carbamazepine	
	F	58	11	5	T	D	yes	clonidine			
	M	23	12	7	TM	C	yes	clonidine			
	F	35	13	3	TM	C	yes	baclofen	yes	amitriptyline	
	F	32	15	4	V	D	yes	baclofen	yes	amitriptyline	
	F	45	16	9	T	D	yes	baclofen			
	F	42	17	8	T	D	yes	baclofen	yes	amitriptyline	
	60% F 40% M	38 ± 9	10 ± 4		27% TM 53% T 20% V	40% C 60% D	73%		67%		
RAR − NIBS	F	65	6	6	T	D	yes	no medication	yes	amitriptyline	
	F	53	7	6	TM	C	yes	tizanidine	yes	amitriptyline	
	M	38	7	5	V	C					
	M	26	7	10	V	D			yes	carbamazepine	
	M	29	9	6	T	C					
	F	35	9	3	TM	C	yes	baclofen	yes	carbamazepine	
	M	35	10	3	TM	C	yes	clonidine	yes	carbamazepine	
	F	51	10	3	T	C	yes	clonidine			
	F	28	10	9	V	D			yes	no medication	
	M	61	10	10	V	D			yes		
	F	64	12	4	T	D	yes	no medication			
	M	45	12	3	T	C	yes	baclofen	yes	no medication	
	F	43	12	5	T	D	yes	tizanidine	yes	gabapentin	
	F	53	13	6	TM	D	yes	tizanidine	yes	gabapentin	
	M	18	13	8	T	C					
	F	48	13	9	T	D					
	M	23	13	6	V	D	yes	baclofen	yes	amitriptyline	
	M	22	14	6	TM	C	yes	clonidine	yes	gabapentin	
	M	42	14	5	T	D	yes	tizanidine	yes	paracetamol	
	F	39	15	7	V	C	yes	tizanidine			
	F	49	15	4	TM	D	yes	baclofen	yes	amitriptyline	
	F	60	16	8	V	D	yes	no medication	yes	amitriptyline	
	F	62	16	7	V	D	yes	no medication			
	F	57	17	8	V	C	yes	baclofen			
	M	55	18	8	T	C	yes	baclofen	yes	paracetamol	
summary	56% F 44% M	44 ± 14	12 ± 3		24% TM 40% T 36% V	48% C 52% D	72%		64%		
*p*-value	0.4	0.1	0.1	0.2	0.2	0.3	0.5		0.4		

Legend: RAR, robot-aided rehabilitation; NIBS, non-invasive brain stimulation; AIS, ASIA Impairment Scale; C and D, AIS grades at study entry; F, female; M, male; m, months; SCI, spinal cord injury; T, trauma; TM, transverse myelitis; V, vascular; y years; TSO, time from SCI onset. p-value of between-group comparison.

**Table 2 brainsci-12-00119-t002:** Outcome measures and statistical data.

		T0	T1	T2	ANCOVA		Between-Group		Within-Group	
					F	*p*	T1–T0	T2–T0	T1–T0	T2–T0
10MWT	RAR + NIBS	0.75 ± 0.49	0.98 ± 0.58	0.8 ± 0.53	10	0.002	<0.0001	0.003	<0.0001	0.01
	RAR − NIBS	0.65 ± 0.35	0.72 ± 0.39	0.78 ± 0.36					<0.0001	<0.0001
6MWT	RAR + NIBS	206 ± 15	248 ± 21	227 ± 24	0.3				<0.0001	<0.0001
	RAR − NIBS	212 ± 16	235 ± 23	224 ± 17					<0.0001	<0.0001
FIM-L	RAR + NIBS	3 ± 1	4 ± 1	4 ± 1	5	0.02	0.01	0.01	<0.0001	<0.0001
	RAR − NIBS	3 ± 1	3 ± 1	3 ± 1					<0.0001	<0.0001
LEMS	RAR + NIBS	31 ± 7	37 ± 8	34 ± 8	9	0.005	<0.0001	0.002	<0.0001	<0.0001
	RAR − NIBS	30 ± 6	33 ± 7	31 ± 6					<0.0001	<0.0001
MAS	RAR + NIBS	1.3 ± 1	1 ± 0.7	1.1 ± 0.8	10	0.002	<0.0001	<0.0001	0.0006	0.001
	RAR − NIBS	1.4 ± 1.1	1.2 ± 1	1.3 ± 1.1					<0.0001	0.0002
TUG	RAR + NIBS	62 ± 27	51 ± 22	55 ± 23	0.5				<0.0001	0.0002
	RAR − NIBS	64 ± 26	57 ± 25	60 ± 24					<0.0001	<0.0001
VAS	RAR + NIBS	3 ± 2	2 ± 2	3 ± 2	12	0.0009	<0.0001	<0.0001	0.001	0.0009
	RAR − NIBS	3 ± 3	3 ± 3	3 ± 3					0.0002	0.0006
WISCI II	RAR + NIBS	8 ± 4	9 ± 5	9 ± 5	10	0.002	<0.0001	<0.0001	0.0004	0.0001
	RAR − NIBS	6 ± 4	7 ± 4	7 ± 4					<0.0001	<0.0001

Legend: robot-aided rehabilitation (RAR), non-invasive brain stimulation (NIBS), 6 min walk test (6MWT), 10 m walk test (10MWT), timed up and go (TUG), Walking Index for Spinal Cord Injury (WISCI II), Functional Independence Measure-Locomotion (FIM-L), lower extremity motor score (LEMS), modified Ashworth scale (MAS), visual analog scale for pain (VAS), Analysis of Covariance (ANCOVA), F-value (F), *p*-value (p).

## Data Availability

The data presented in this study are available upon request from the corresponding author.

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
