# Peer review of "Finding the Way to Improve Motor Recovery of Patients with Spinal Cord Lesions: A Case-Control Pilot Study on a Novel Neuromodulation Approach"

_brainsci, 2022, doi:10.3390/brainsci12010119_

Round 1
Reviewer 1 Report
This is an interesting study examining the effect of combined noninvasive brain stimulation and robot-assisted gait training in patients with incomplete spinal cord injury.
Because there are still limited therapeutic approaches for improving motor and gait function of the CNS injury patients, this novel approach combining central (brain and spinal cord) and peripheral (robot-assisted training) neuromodulation methods can be suggestive of one of practical interventions for these patients.
However, this study has major limitation of not performing randomized controlled trial. Instead, the authors used propensity matching analysis of the subjects which had limited strength of beneficial effect of the experimental approach.
Nevertheless, this novel approach has some value of pilot trial for examining safety and feasibility of combined neuromodulation approaches for CNS injury patients and gives insight for further study in this field.
I would suggest to the authors to clarify the limitation of the study design and carefully interpret the results of this study. The title should have a "Pilot Study".
Most of participants are taking medications. Authors should describe how their medication status were controlled. If there were any changes of medication during the experimental period, it should be described in detail.
Author Response
To the Reviewer#1
- This is an interesting study examining the effect of combined noninvasive brain stimulation and robot-assisted gait training in patients with incomplete spinal cord injury. Because there are still limited therapeutic approaches for improving motor and gait function of the CNS injury patients, this novel approach combining central (brain and spinal cord) and peripheral (robot-assisted training) neuromodulation methods can be suggestive of one of practical interventions for these patients. However, this study has major limitation of not performing randomized controlled trial. Instead, the authors used propensity matching analysis of the subjects which had limited strength of beneficial effect of the experimental approach. Nevertheless, this novel approach has some value of pilot trial for examining safety and feasibility of combined neuromodulation approaches for CNS injury patients and gives insight for further study in this field. I would suggest to the authors to clarify the limitation of the study design and carefully interpret the results of this study. The title should have a "Pilot Study".
We want to thank you for your appreciation of our ms and the helpful suggestions to improve its quality. We agree with the reviewer’s point of view. So, we clarified the limitation of the study design (see Par. 4.1), thus consistently revising the study conclusions. “Pilot study” was added to the title as suggested.
- Most of participants are taking medications. Authors should describe how their medication status were controlled. If there were any changes of medication during the experimental period, it should be described in detail.
All participants were hospitalized at the Neurorobotic Unit of our institute, so that patients’ medication status was easily controlled. Further, patients were not given any medication change during the experimental period. Therefore, this paragraph was added to the text, as suggested.
Kindest regards,
The authors.
Reviewer 2 Report
The control group should be mentioned in the abstract since it is indicated "both groups"
Author Response
To the Reviewer#2
The control group should be mentioned in the abstract since it is indicated "both groups".
We want to thank you for your appreciation of our ms and the helpful suggestions to improve its quality. The ms was revised to improve its readability. The suggested correction was applied to the text.
Kindest regards,
The authors.